# Climate, ocean circulation, and sea level changes under stabilization and overshoot pathways to 1.5K warming

Jaime B. Palter[1], Thomas L. Frölicher[2,3], David Paynter[4], and Jasmin G. John[4]

[1]University of Rhode Island, Graduate School of Oceanography
[2]Climate and Environmental Physics, Physics Institute, University of Bern, Switzerland
[3]Oeschger Centre for Climate Change Research, University of Bern, Switzerland
[4]National Oceanic and Atmospheric Administration, Geophysical Fluid Dynamics Laboratory

*Correspondence to:* Jaime B. Palter (jpalter@uri.edu)

**Abstract.** The Paris Climate Agreement has initiated a scientific debate on the role that carbon removal - or net negative emissions - might play in achieving less than 1.5K of global mean surface warming by 2100. Here, we probe the sensitivity of a comprehensive Earth System Model (GFDL-ESM2M) to three different atmospheric $CO_2$ concentration pathways, two of which arrive at 1.5K of warming in 2100 by very different pathways. We run five ensemble members of each of these
5   simulations: 1) a standard Representative Concentration Pathway (RCP4.5) scenario, which produces 2K of surface warming by 2100 in our model; 2) a 'stabilization' pathway in which atmospheric $CO_2$ concentration never exceeds 440 ppm and the global mean temperature rise is approximately 1.5K by 2100; and 3) an 'overshoot' pathway that passes through 2K of warming at mid-century, before ramping down atmospheric $CO_2$ concentrations, as if using carbon removal, to end at 1.5K of warming at 2100. Although the global mean surface temperature change in response to the overshoot pathway is similar
10   to the stabilization pathway in 2100, this similarity belies several important differences in other climate metrics, such as warming over land masses, the strength of the Atlantic Meridional Overturning Circulation (AMOC), ocean acidification, sea ice coverage, and the global mean sea level change and its regional expressions. In 2100, the 'overshoot' ensemble shows a greater global steric sea level rise and weaker AMOC mass transport than in the stabilization scenario, with both of these metrics close to the ensemble mean of RCP4.5. There is strong ocean surface cooling in the North Atlantic and Southern Ocean
15   in response to overshoot forcing due to perturbations in the ocean circulation. Thus, overshoot forcing in this model reduces the rate of sea ice loss in the Labrador, Nordic, Ross, and Weddell Seas relative to the stabilized pathway, suggesting a negative radiative feedback in response to the early rapid warming. Finally, the ocean perturbation in response to warming leads to strong pathway-dependence of sea level rise in northern North American cities, with overshoot forcing producing up to 10 cm of additional sea level rise by 2100 relative to stabilization forcing.

# 1 Introduction

In late 2015, the Paris Climate Accord promoted the goal of keeping a global temperature rise to well below 2K above prein-dustrial levels, and to pursue further efforts to limit the temperature increase to 1.5K (http://unfccc.int). However, by 2015, our planet's surface temperature had already surpassed 1K above the preindustrial average, and current emission mitigation policies are very unlikely to limit warming to less than 1.5K by 2100 (Raftery et al., 2017). Even under optimistic scenarios of a downturn in the global emissions rate, avoiding global warming in excess of the 1.5K threshold may require a period of net negative emissions – i.e. the on-site capture of $CO_2$ at emission sources and the removal of $CO_2$ from the atmosphere (Gasser et al., 2015; Sanderson et al., 2016).

Presently, the details of the climate response to different emissions pathways to 1.5K, including a period with net negative emissions, are unclear. So far, reversibility studies that considered carbon removal have principally relied on highly idealized pathways with a limited number of models, and have focused principally on global mean quantities with little attention to regional climate effects (Boucher et al., 2012; Tokarska and Zickfeld, 2015; Jones et al., 2016; Zickfeld et al., 2016; Jackson et al., 2014). Moreover, recent results connecting ocean circulation, ocean heat uptake, and cloud feedbacks (Rugenstein et al., 2012; Winton et al., 2013; Galbraith et al., 2016; Trossman et al., 2016) have revealed the potential for nonlinearities and hysteresis in the climate system that could make the rate of transient warming and its regional expressions pathway-dependent.

The canonical example of such nonlinearity and/or hysteresis in the climate system is the response of the Atlantic Meridional Overturning Circulation (AMOC) to warming. AMOC influences global climate through its impact on ocean heat uptake (Kostov et al., 2014; Winton et al., 2013) and the meridional transport of heat and associated impacts on radiative feedbacks (Herweijer et al., 2005; Zhang et al., 2010; Trossman et al., 2016). AMOC change may alter the global feedback parameter in a time-dependent way, for instance, by altering the rate of sea ice loss in the Arctic or mid-latitude cloud cover, with the net result difficult to anticipate (Zhang et al., 2010; Rugenstein et al., 2016; Trossman et al., 2016).

In this study we use a set of three simulations, each with five ensemble members, conducted with a comprehensive Earth System Model (ESM) to explore how different $CO_2$ concentration pathways influence ocean circulation, sea-ice cover, and sea level rise. We first conduct standard RCP4.5 simulations, a scenario intended to produce 4.5 W m$^{-2}$ of radiative forcing by 2100 (Moss et al., 2010; van Vuuren et al., 2011). Our model realizes approximately 2K of warming by 2100 under the RCP4.5 forcing. These simulations are compared to idealized 'stabilization' pathway simulations where global warming is held below 1.5K by reducing the rate of increase in atmospheric $CO_2$ concentrations starting in 2018, so as to only reach 440 ppm $CO_2$ by 2100 (Figure 1a). The RCP4.5 scenario simulations are also compared to idealized 'overshoot' simulations in which atmospheric $CO_2$ rises as it would under a business-as-usual scenario (RCP8.5) until mid-century, after which it declines rapidly during a period meant to emulate net negative emissions (Figure 1a). At 2100, the overshoot and stabilization pathways have the same radiative forcing.

Our goal in setting up the two new pathways (i.e. 'stabilization' and 'overshoot') was to limit global mean surface warming to 1.5K by 2100 by altering atmospheric $CO_2$ concentrations in drastically different manners that reflect two extremes of hypothetical short-term and long-term climate policy. There are several other potential overshoot and stabilization pathways

that could limit global surface warming to 1.5K over various time periods (Sanderson et al., 2016; Zickfeld et al., 2016; Gasser et al., 2015). We describe the model set up and analysis methods in Section 2, before presenting the resulting global average picture in Section 3. In Section 4, we narrow our focus to differences in regional expressions of climate change that are associated with ocean circulation perturbations. Finally, we conclude and offer an outlook for the future in Section 5.

## 2   Methods

### 2.1   Model and simulation description

All simulations use the NOAA-GFDL Earth System Model, GFDL-ESM2M (Dunne et al., 2012, 2013), which closes the carbon cycle. The ocean component is on 50 fixed depth levels at a nominal $1^o$ resolution, and is coupled to an ocean biogeo-chemical model, TOPAZ2 (Dunne et al., 2013). The atmospheric model of GFDL-ESM2M has a $2^o$x$2.5^o$ (latitude x longitude) horizontal resolution with 24 vertical levels and has identical atmospheric parameterizations to those in GFDL AM2.1 (Anderson et al., 2004). The land model is LM3.0, which represents land, water, energy and carbon cycles. The model is coupled to a dynamic-thermodynamic sea ice model as described in Winton (2000).

Our preindustrial (PI) control simulation was taken from the last 300 years of a simulation that had already been spun up for over 2000 years. As such, there is extremely small drift in global average ocean temperature. Our main ensemble simulations with distinct future pathways were branched from the end of an existing historical simulation (1861-2005), which was conducted using the Coupled Model Intercomparison Project, Phase 5 (CMIP5) protocols. Each ensemble consists of five members with perturbed oceanic initial states but with the same atmospheric, land, ocean biogeochemical, sea-ice, and icebergs initial conditions. Specifically, for each ensemble member, $i=1,2...5$, we added a small temperature perturbation, $dT = 0.0001$ K*$i$, to a single ocean grid cell in the Weddell Sea at 5 m depth, similar to the approach by Wittenberg et al. (2014). Five ensemble members provide a means of averaging out internal variability in order to more clearly separate the differences in the simulations arising from the $CO_2$ forcing at a reasonable computational expense.

The first pathway is a standard RCP4.5 simulation, which in GFDL-ESM2M leads to global average surface warming of 1.96 ± 0.12K (Figure 1c), averaged (±1 standard deviation) over all 5-ensemble members for the final year (2100) of the simulation. Second, a 'stabilization' pathway was created and designed to limit global mean warming to 1.5K above the preindustrial control by 2100. This simulation suite leads to global warming in 2100 of 1.52 ± 0.16K. This warming target was achieved by setting atmospheric $CO_2$ growth rates to be approximately 0.65 ppm year$^{-1}$ from 2020 to 2060, and then having these growth rates decline to nearly zero by the end of the century. Under these slow growth rates, atmospheric $CO_2$ concentrations never exceed 435 ppm before 2100 (Figure 1a). Finally, the 'overshoot' simulation prescribes atmospheric $CO_2$ concentrations equal to those used in the RCP8.5 scenario until 2060, reaching a peak of 573 ppm, after which time $CO_2$ declines rapidly and linearly to 438 ppm in 2100 (Figure 1a). The result is a temperature rise of 1.54 ± 0.14K in 2100 in this simulation suite. For both the overshoot and stabilization pathways, we alter only $CO_2$ concentrations - all other forcing agents between 2006 and 2100 are as prescribed in the standard RCP4.5 pathway. Here, we give the ensemble mean temperature in 2100, since this is the year that the $CO_2$ forcing is approximately equal in the stabilization and overshoot simulations. The ensemble averages over

the years 2096-2100, which are reflected in the final year of the smoothed time series in Figure 1c, are slightly different (1.95 ± 0.05K in RCP4.5; 1.48 ± 0.09K in stabilization; and 1.56 ± 0.09K in overshoot).

To create our chosen pathways, we needed the ability to estimate the temperature response of GFDL-ESM2M to changing $CO_2$ forcing before performing model runs. In the stabilization case we needed to estimate how much the steady $CO_2$ increase

in RCP4.5 would need to be reduced to limit warming to below 1.5K rather than 2.0K of global warming by 2100. In the overshoot case, we estimated the year in which the model would reach 2K following the RCP8.5 $CO_2$ concentrations, and what decrease in $CO_2$ would be required to reach 1.5K in 2100. To achieve this goal we made use of the time-series of global mean temperature in a 4500-year run of GFDL-ESM2M under 1% per year $CO_2$ increase from preindustrial levels to year 70 (year of doubling), after which atmospheric $CO_2$ is held constant (Paynter et al., 2018). Global mean surface air temperature, $T$, in

some year $t$, relative to an initial year, $t_0$ (assumed here 1860), can be estimated using the following scaling of the temperature time-series in response to a doubling of $CO_2$:

$$T(t) = \sum_{i=t_0}^{t} \left( \frac{\Delta F(i)}{F_{2xCO2}} \right) T_{2xCO2}(t-i) \tag{1}$$

where $\Delta F(i)$ is the incremental change in forcing in a given year $i$, $F_{2xCO2}$ is the radiative forcing in response to a doubling of $CO_2$, and $T_{2xCO2}$ is the time series of the global mean surface air temperature response in GFDL-ESM2M to a doubling of $CO_2$ (see Supplementary Figure S1 for a comparison of the results of this scaling to the GFDL-ESM2M global mean surface

temperatures). Forcing is estimated using the formulas in Etminan et al. (2016).

Using this simple technique, we were able to estimate the 2006-2100 temperature response of the full GFDL-ESM2M model forced with the RCP4.5 and RCP8.5 pathways within ±0.2K. Our estimation method assumes that both the stabilization and overshoot pathways would require the same atmospheric concentration of $CO_2$ to be below 1.5K at 2100, suggesting that hitting the 1.5K global average target is, to first order, independent of the $CO_2$ concentration pathway. This result was largely

confirmed by running GFDL-ESM2M under the stabilization and overshoot pathways. The surface warming by 2100 in the overshoot pathway (1.54 ± 0.14K) was not statistically different than the stabilization pathway (1.52 ± 0.16K). However, as Sections 3 and 4 demonstrate, this similar global mean surface air temperature actually results from quite different climate states.

## 2.2   Analysis Methods

We calculate the AMOC stream function as the vertically- and zonally-integrated meridional transport in the Atlantic basin:

$$\Psi(y,z) = \int_z^{\eta} \int_{x_w}^{x_e} v \, dx \, dz \tag{2}$$

where $v$ is the meridional velocity due to the sum of resolved and parameterized mesoscale and submesoscale processes, $x$ is the zonal coordinate increasing eastward, $z$ is the vertical coordinate increasing upward, $\eta$ is the ocean free surface, and

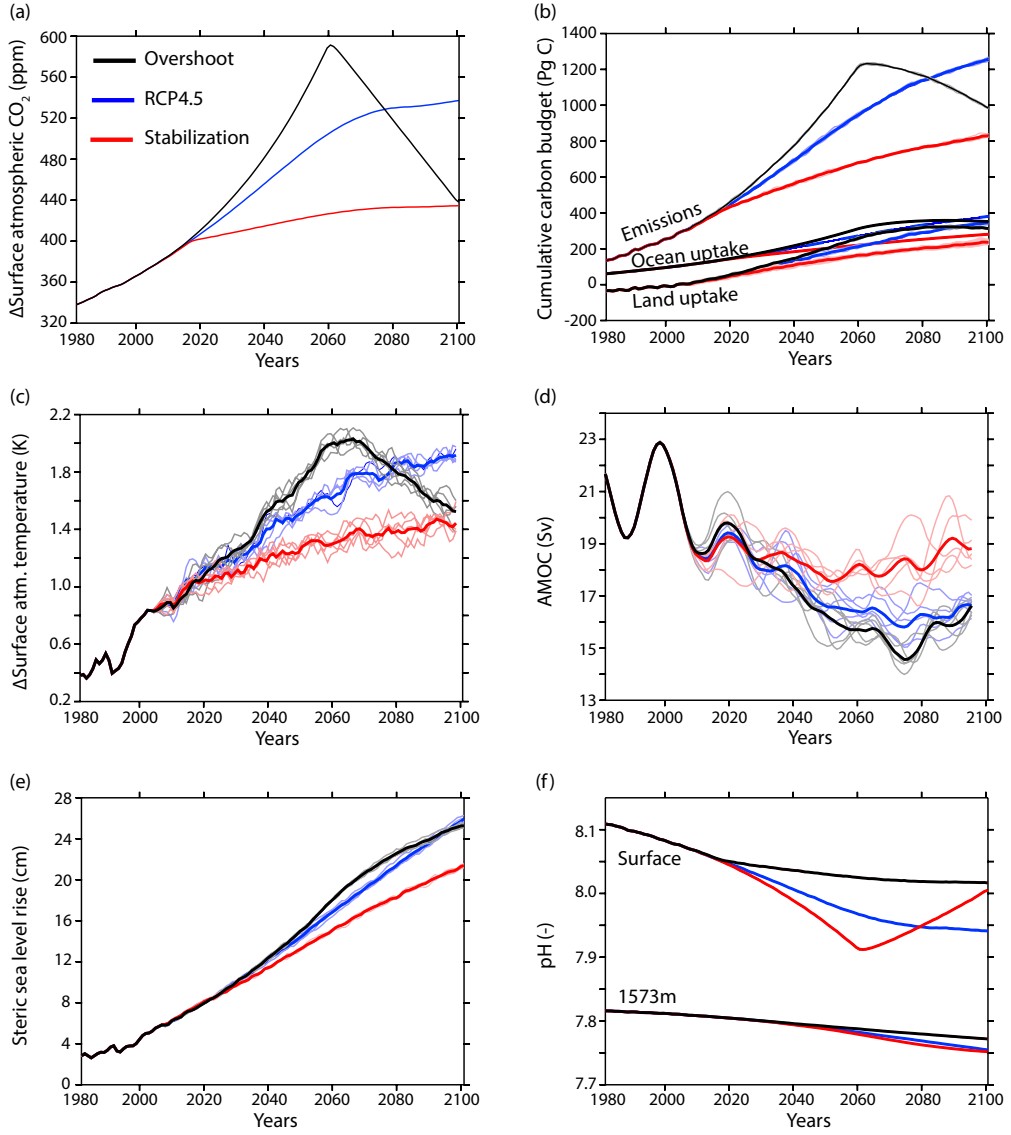

**Figure 1.** Time series of global average properties of the three simulations: RCP4.5 (blue), stabilization (red), and overshoot (black). The last 25 years of the historical simulation (1980-2005) are also shown in black. a) Surface atmospheric $CO_2$ (ppm). b) The cumulative carbon budget (PgC). c) The change in surface atmospheric temperature relative to the preindustrial control simulation (K). d) The Atlantic Meridional Overturning Circulation, defined in this figure as the vertical maximum of the zonally- and vertically-integrated meridional velocity in the North Atlantic Basin at $42^oN$ (Sv). e) Global steric sea level rise (cm). See Methods for calculating carbon budget and for the equation relating steric sea level change to in situ density. In panels b-e, ensemble members are shown as thin lines and the ensemble average is shown as a thick line. Time series in (c) and (d) are smoothed with a five and ten year running mean, respectively.

$x_w$ and $x_e$ are the westward and eastward boundaries of the North Atlantic basin. The simulated AMOC of 17.0 Sv at $26^o$N averaged over the period 1996 to 2015 in the RCP4.5 scenario is close to the mean of the AMOC measured by the UK-led RAPID project (17.2 Sv), which has monitored the overturning circulation with an array of moorings at this latitude since 2004 (Cunningham et al., 2007). Hereafter, we report on the annual mean of the vertical maximum of that transport at $42^o$N, since one area of focus is on subpolar temperatures and sea level. The temporal evolution of the maximum overturning stream function over the historical period and the projected $21^{st}$ century is qualitatively similar for each of the scenarios regardless of the latitude where it is computed.

It is useful to understand how sea level tendencies may be separated into tendencies from mass and local steric changes (Landerer et al., 2007; Griffies et al., 2014):

$$\frac{\partial \eta}{\partial t} = \frac{1}{g\rho_0}\Big(\frac{\partial(p_b - p_a)}{\partial t}\Big) - \frac{1}{\rho_0}\int\limits_{-H}^{\eta}\frac{d\rho}{dt}dz \tag{3}$$

where $\eta$ is the sea level, $p_b$ is the bottom pressure and $p_a$ is the surface pressure, $\rho$ is the *in situ* density, and $\rho_0$ is a reference density. The first term on the right hand side of equation 3 is due to the change in mass of a fluid column, either through the divergence of the vertically integrated horizontal currents or a mass flux across its boundary. In this model, as with most climate model simulations, atmospheric pressure loading is neglected, so $p_a$ is zero except due to mass loading under sea ice (Griffies et al., 2014). The second term on the right hand side of equation 3 is the local steric sea level term, which is dominated by thermal expansion under our global warming simulations. In Boussinesq ocean models like the one used here, the global average steric sea level change must be diagnosed from the in situ density field and added to the local steric effect, which is prognosed in the model. For a detailed exploration of physical controls on sea level in ocean models, including the implementation of sea level budgets in a Boussinesq model, see Griffies and Greatbatch (2012).

It is crucial to note that changes in the global average mass of the ocean in our simulations arise only from changes in the balance of precipitation, evaporation and river inputs, as our model does not represent interactive ice sheets or glaciers. However, melting ice sheets and glaciers are projected to contribute more than half of the global average sea level rise by 2100 under various forcing scenarios (Stocker et al., 2013). Thus, our global average sea level rise plots likely represent less than half the total expected sea level rise at the end of the $21^{st}$ century. Moreover, Greenland's melting ice sheet may contribute to freshening of the subpolar North Atlantic, which could hasten and/or intensify an AMOC slowdown. Finally, our model also ignores changes in local sea level due to changes in land elevation, gravitational and isostatic effects, as well as changes in atmospheric pressure loading, known as the inverse barometer effect (Griffies et al., 2014).

## 3 Global Average Properties

The three simulation suites, which differ only in their prescribed atmospheric $CO_2$ concentrations (Figure 1a), branch into distinctive futures from the conclusion of the historical period to the end of the century (2006-2100). It is important to note that the GFDL-ESM2M transient climate response (TCR) is 1.3K, making it one of the smallest among the CMIP5 models (Gillett

et al., 2013; Flato et al., 2013). Therefore, the allowable carbon emissions to limit warming to 2K or 1.5K is larger than that estimated by most CMIP5 models (Frölicher and Paynter, 2015). Nevertheless, it is instructive to compare the diagnosed cumulative carbon emissions that are compatible with the prescribed atmospheric $CO_2$ concentration pathways of our overshoot, stabilization and RCP4.5 scenarios (Figure 1b). The diagnosed cumulative carbon emissions from 1861-2100 (relative to 1861-1880) are 830 GtC in the stabilization pathway. In the overshoot simulations, diagnosed cumulative emissions peak at 1232 GtC in year 2062, similar to the RCP4.5 cumulative emissions in 2100 (1257 PgC). Yet, to achieve almost equal atmospheric $CO_2$ concentration forcing as the stabilization pathway in year 2100, "only" 248 GtC must be removed (6.5 GtC year$^{-1}$). The net extra 154 GtC hypothetically emitted in the overshoot scenario is stored in the ocean and terrestrial biospheres, which provide a carbon sink that increases at higher levels of atmospheric $CO_2$, a response previously noted for an intermediate complexity Earth System Model with prescribed emissions (Tokarska and Zickfeld, 2015). Indeed, in the overshoot simulations, penetration of carbon into the ocean interior and the land biosphere permits the ocean and the land to remain net carbon sinks until year 2087 and year 2086, respectively (Figure 1b). From those years to the end of the 21$^{st}$ century, the ocean and the land carbon inventory decline by 6.6 and 9.8 PgC.

The higher atmospheric carbon concentrations in the overshoot pathway relative to the stabilization pathway temporarily cause more ocean acidification (i.e. lower pH values) at the surface (Fig. 1f). We expect this result to be robust to the model choice, since there is little difference in surface pH among the CMIP5 models (Frölicher et al., 2016). In 2100, when the atmospheric $CO_2$ in overshoot nearly matches stabilization, the surface pH is still lower in overshoot, but is rapidly approaching the stabilization surface value. At depth (e.g. 1573 m), both the emergence of greater acidification as well as the turn-around in pH in the overshoot scenario are delayed by about 20 years relative to the trends in atmospheric $CO_2$, as transport and mixing between the surface and interior ocean is relatively slow (Gebbie et al., 2012). As a result, ocean pH is lower in the overshoot scenario than in the stabilization scenario at the end of the 21st century, as expected from the carbon budget (Figure 1b). Based on the rate of change at 2100 in our simulations, we expect this difference in pH to persist for a few decades after the crossover point in atmospheric $CO_2$, as the ocean slowly releases the excess $CO_2$ to the atmosphere.

The overshoot and stabilization simulations end with approximately the same atmospheric $CO_2$ concentration in 2100, and the ensemble average global mean surface air temperatures at 2100 are statistically indistinguishable (Figure 1a,c). Thus, for global average surface air temperature, the pathway to a given forcing does not factor prominently in the annual mean, global average surface air temperature change in response to that forcing. Because the hydrological cycle tends to spin-up as the climate warms (Held et al., 2006), the global average precipitation increases approximately 1% over the 21st century in all three scenarios. The rate of change seems to roughly track the global temperature change, therefore leading to slightly stronger increases in Overshoot and RCP4.5 starting at mid-century (Supplementary Figure 2). However, internal variability causes substantial overlap between ensemble members of the three scenarios.

On the other hand, there are very important differences in other properties between every member of the overshoot and stabilization ensembles. For instance, steric sea level rise reflects the total ocean heat content change over the entire simulation, and to first order is due to the time-integral of the top of the atmosphere (TOA) radiative imbalance (Trenberth et al., 2014). The overshoot simulation experiences global average steric sea level rise 18% higher (or 3.9 cm) than the stabilization pathway,

and is essentially identical to the steric sea level rise under RCP4.5 forcing in 2100 (Figure 1e). Thus, as discussed in previous papers (e.g. Tokarska and Zickfeld, 2015), the overshoot pathway has long-term and difficult-to-reverse consequences for sea level rise, given that some of this heat enters the deep ocean and will take centuries to equilibrate (Drijfhout et al., 2012; Swingedouw et al., 2008).

5     Ocean heat uptake per unit area in the Atlantic outpaces the other basins (Supplementary Figure S3), with this pattern arising in part due to the overturning circulation and its slowdown (Winton et al., 2013). Notably, the Atlantic heat uptake in RCP4.5 and Overshoot are slightly faster than the uptake in Stabilization, even after removing the global mean uptake rate.

    The AMOC change is also strongly connected to forcing pathway. Before branching off the three simulation suites, the AMOC had already slowed by 11% over the historical period (the average over 1986-2005 compared to 1861-1880), from 10   23.7 to 21.1 Sv at $42^o$N. The AMOC shows no further decline in the stabilization suite in the $21^{st}$ century. In contrast, in both the RCP4.5 and overshoot simulation suites, the AMOC slowed by an additional 23% by the end of the $21^{st}$ century (Figure 1d). The AMOC variability mirrors the radiative forcing with an approximately 15-year lag in response to overshoot forcing: atmospheric $CO_2$ peaks in 2060 and declines thereafter (Figure 1a), whereas the AMOC reaches its lowest point in 2075 (the ten year mean centered on 2075 is 14.6 Sv), after which it begins climbing towards recovery. By 2100, the AMOC remains 15   2.5 Sv slower in the overshoot pathway relative to the stabilization pathway, and about equal to the AMOC under RCP4.5 forcing. In comparison to 22 other CMIP5 models, ESM2M was shown to have the seventh strongest AMOC circulation under historical boundary conditions (Wang et al., 2014). A strong base state of the AMOC is known to be a predictor for a large AMOC decline (Winton et al., 2014), and the AMOC slowdown in under RCP4.5 forcing is, indeed, among the stronger responses of the CMIP5 models (Cheng et al., 2013).

20  **4   Regional patterns tied to the ocean circulation perturbation**

    As is common for climate models responding to anthropogenic climate change (Stocker et al., 2013), the surface air temperature at the end of the $21^{st}$ century in all three scenarios is warmest in the Arctic, coolest over the Southern Ocean, with a global minimum in surface temperature change over the subpolar North Atlantic, the so-called "warming hole" (see Figure 2a for the stabilization scenario). In the stabilization scenario, the subpolar North Atlantic warming hole region (averaged from 50°W-25   30°W, 47°N-60°N) is 1.77K cooler over the period 2096-2100 than in the preindustrial; the RCP4.5 scenario is 1.22K cooler in that region than the preindustrial (Figure 2b). Surface salinity in this region declines in all three scenarios, from an average of 34.7 g kg$^{-1}$ from 2006-2011 to 34.13 ± 0.07 in Overshoot, 34.35 ± 0.02 in Stabilization, and 34.25 ± 0.03 in RCP4.5 from 2096-2100 (where the error is one ensemble standard deviation). This freshening occurs despite the absence of interactive ice sheets. A fresher surface layer increases the buoyancy forcing required to overturn the water column and sustain the sinking 30   branch of the AMOC.

    Heat budget terms for the ocean surface layer are an ideal tool to examine the mechanisms governing temperature within the "warming hole" in the subpolar North Atlantic. Tendency terms for each process that affects ocean temperature are available in GFDL-ESM2M for perfect closure of a heat budget. Figure 3 shows the leading heat budget terms for the last 95-years of the

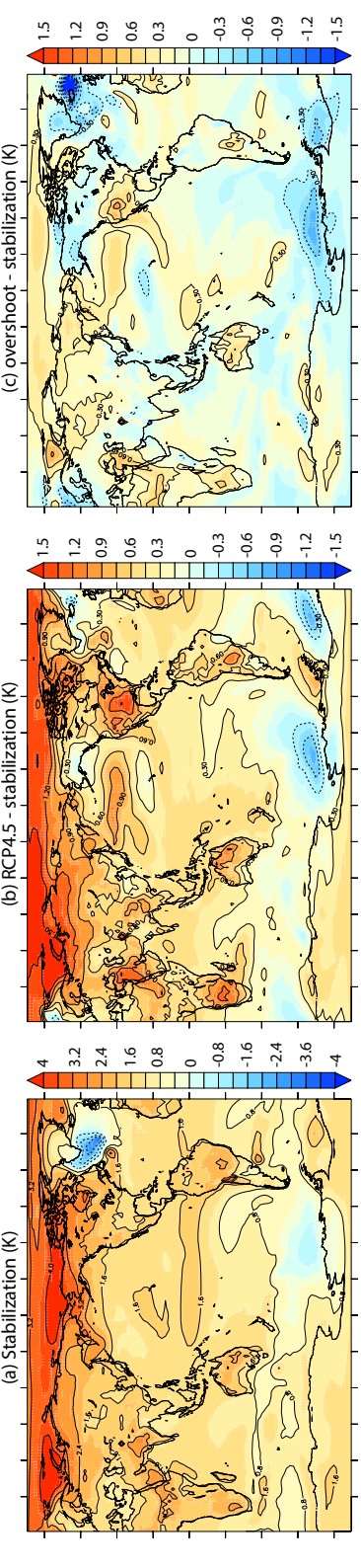

**Figure 2.** Surface temperature change at the end of the century (K). a) Stabilization pathway minus preindustrial control average, b) RCP4.5 scenario minus the stabilization pathway, c) overshoot pathway minus stabilization pathway. All maps are made from the ensemble average, also averaged over the years 2096-2100.

preindustrial control, and their difference from that control in the stabilization and overshoot simulations, vertically averaged over the top 100 m, for the North Atlantic region (60°W – 0, 48-65°N) and temporally-integrated over the period 2006-2100. The RCP4.5 heat budget perturbation (not shown) is qualitatively similar to the stabilization and overshoot simulations. The preindustrial control simulation is very nearly in equilibrium due to a balance between heat supplied by oceanic advection and

mixing on neutral surfaces and removed by exchange with the atmosphere. Between 2006 and 2100, the stabilization scenario sees a slowdown in the advective supply of heat to this layer by 76%, and the overshoot scenario by 84%. It is interesting that this slowdown in the advective supply of heat is apparent in the stabilization ensemble despite the fact that its AMOC volume transport is essentially unchanging between 2006 and 2100, and saw only an 11% decline during the historical years. The large slowdown in advective heat supply may suggest a role for the horizontal gyre circulations, as was recently shown to be the

case on interannual time scales (Piecuch et al., 2017). Indeed, the spatial pattern of sea level changes are consistent with an overall slackening of gyre circulation, with a negative anomaly in the subtropical gyre and positive anomaly in the subpolar gyre (Figure 4).

The slowdown in the advective supply of heat is associated with a strong weakening of ocean heat loss to the atmosphere. In addition, lateral mixing along isopycnals, which is a small warming term to this layer in the preindustrial, approximately

doubles in the perturbation simulations. The combined warming effect of the weakened air-sea flux and increased lateral mixing of heat is slightly smaller than the slowdown in the advective supply of heat, thereby producing a weak cooling of the subpolar North Atlantic at the surface. The full-depth preindustrial heat budget (not shown) is similar to the 0-100 m heat budget, in that it is dominated by a near balance between the advective supply of heat and the cooling influence of air-sea exchange, both of which weaken in the $21^{st}$ century. However, in contrast to the surface layer, the column-integrated decline in ocean

to atmosphere heat flux and warming effect of neutral mixing are slightly larger than the decline in the advective heat supply, thereby creating a small column-average warming in the stabilization simulation (0.09K), that is slightly larger in the overshoot scenario (0.11K).

The cooling of the North Atlantic surface ocean is associated with a reduced rate of sea ice decline, or even slight expansion in the Labrador, Irminger and Norwegian Seas (Figure 5). Thus, the ocean circulation perturbation appears to have provided a

regional stabilizing feedback by reducing the albedo decline associated with sea ice melting.

The surface of the Atlantic and Pacific sectors of the Southern Ocean are also considerably cooler in the RCP4.5 and overshoot simulations than the stabilization simulations (Figure 2). A co-located decline in the sea level pattern (Figure 4) reveals the cause of change in the Pacific sector: a northward shift in the Antarctic Circumpolar Current (ACC) transports cooler water into this region. A northward shift of the ACC core and southern boundary has been previously documented for

this model under RCP4.5 forcing: Meijers et al. (2012) showed that the zonal mean ACC position in the GFDL-ESM2M model shifts northward by 0.4°, due primarily to a large northward excursion in the Pacific sector. CMIP5 models show a wide variety of behavior with respect to evolving ACC position, with no consensus emerging as to whether the current is likely to shift north or south in response to climate change. Thus, this regional cooling is strongly model dependent. One consequence of the northward shift in the ACC is the expansion of southern hemisphere sea ice (Figure 5), which further cools the region, similar

to the situation in the North Atlantic.

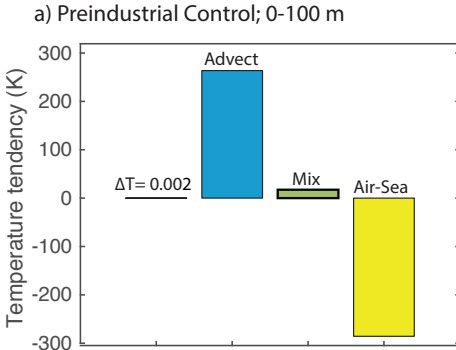
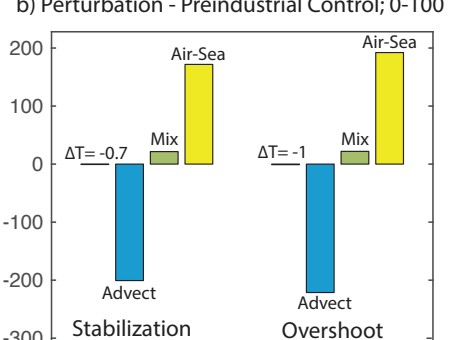

**Figure 3.** Near-surface layer (0-100 m) averaged subpolar North Atlantic heat budget. a) The temporal integral of the last 95 years of the preindustrial control simulation; b) The temporal integral of these terms in the perturbation simulations (years 2006-2100) minus the preindustrial control for (left) stabilization and (right) overshoot pathways. All numbers are ensemble means, averaged with appropriate volume-weighting over the region 60°W – 0 and 48-65°N, from the sea surface to 100 m. The advective term ("advect") is due to the sum of the resolved advection and parameterized mesoscale and submesoscale eddies. "Air-sea" represents the turbulent heat flux between the ocean and atmosphere. "Mixing" is due to diffusion across temperature gradients on neutral surfaces. Only the leading terms in the budget are shown.

The spatial expression of sea level rise is of immediate societal consequence. As noted above, global mean sea level rise in the overshoot pathway responds to the time integral of the TOA radiative imbalance and, as such, is 18% higher than the stabilization pathway by 2100 (Figure 1e). However, ocean circulation changes can either intensify or oppose the global average trend in a given locale. The sea level rise experienced by the heavily populated coastlines on either side of the North Atlantic

5    is extremely sensitive to the pathway to 1.5K. There is a strong dynamic dividing line in the sea level trends at approximately Cape Hatteras ($36^oN$): cities to the north (e.g. Boston and St. John's, Newfoundland, Figure 6a,b) would experience up to 10 cm higher sea levels in 2100 under our overshoot pathway relative to the stabilization pathway, whereas cities to the south (Charleston and Miami, Figure 6d,e) would experience less than 4 cm difference between sea level rise under under stabilization and overshoot pathways. Similar patterns (i.e. accelerated sea level rise in northern North American cities) have been described

10    in response to AMOC perturbations, like those produced by either radiative forcing or freshwater hosing simulations in the North Atlantic (Yin et al., 2009). On the other side of the Atlantic, southern Iberian Peninsula and West Africa have slightly accelerated sea level rise under the overshoot pathway, while further north on the western European coast experiences similar sea level rise in response to overshoot and stabilization pathways (Figures 4). In Brest France, for instance, the local sea level

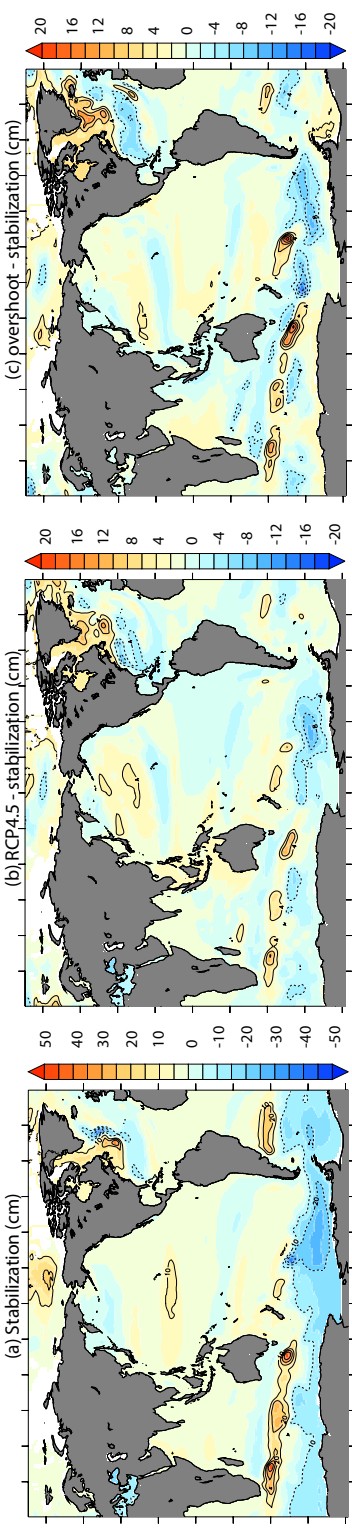

**Figure 4.** Spatial pattern of sea level rise (cm), with the global average steric sea level rise removed. All maps are made from the ensemble average, also averaged over the years 2096-2100. a) Stabilization. a) Stabilization - preindustrial control, b) stabilization - RCP4.5, c) stabilization-overshoot. To calculate the total modeled sea level rise (which still neglects melting ice sheets and glaciers) one would add the global mean steric sea level of 20.2 cm to panel a). The global steric effect averages 24.1 cm in the RCP4.5 simulations and 24.2 cm in the overshoot simulations.

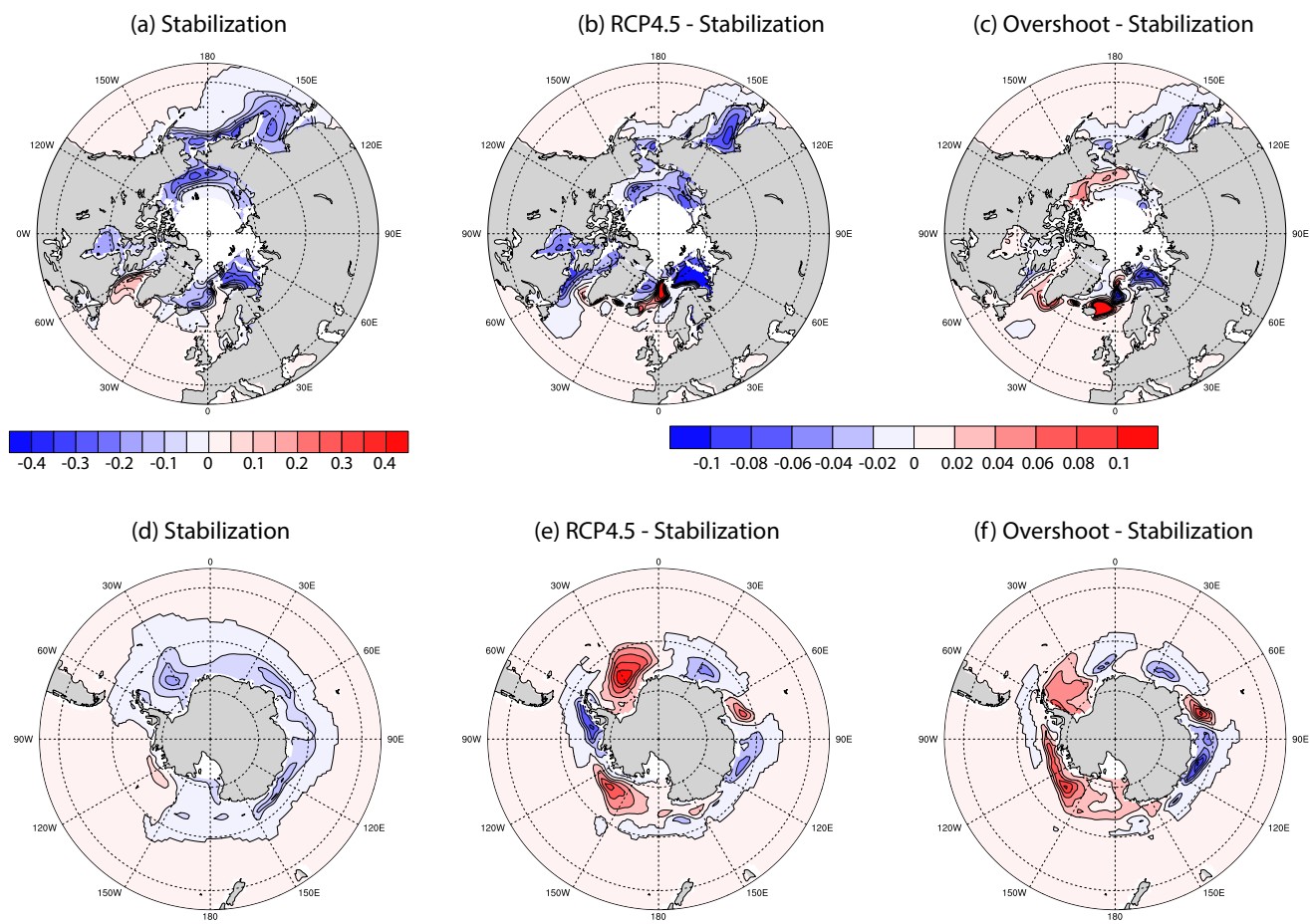

**Figure 5.** The change in sea ice concentration (given as fraction of the grid cell area covered in sea ice). a) Stabilization pathway - preindustrial control, b) RCP4.5 scenario minus the stabilization pathway, c) overshoot pathway minus stabilization pathway. Note that there are 5 vertical layers of sea ice, in which the concentration sums to 1 if the cell is entirely covered with sea ice for the entire year. Therefore, the change is given as the difference in the sum over the five layers of the ensemble mean 2096-2100 average minus the sum averaged over the final 300 years of the preindustrial control.

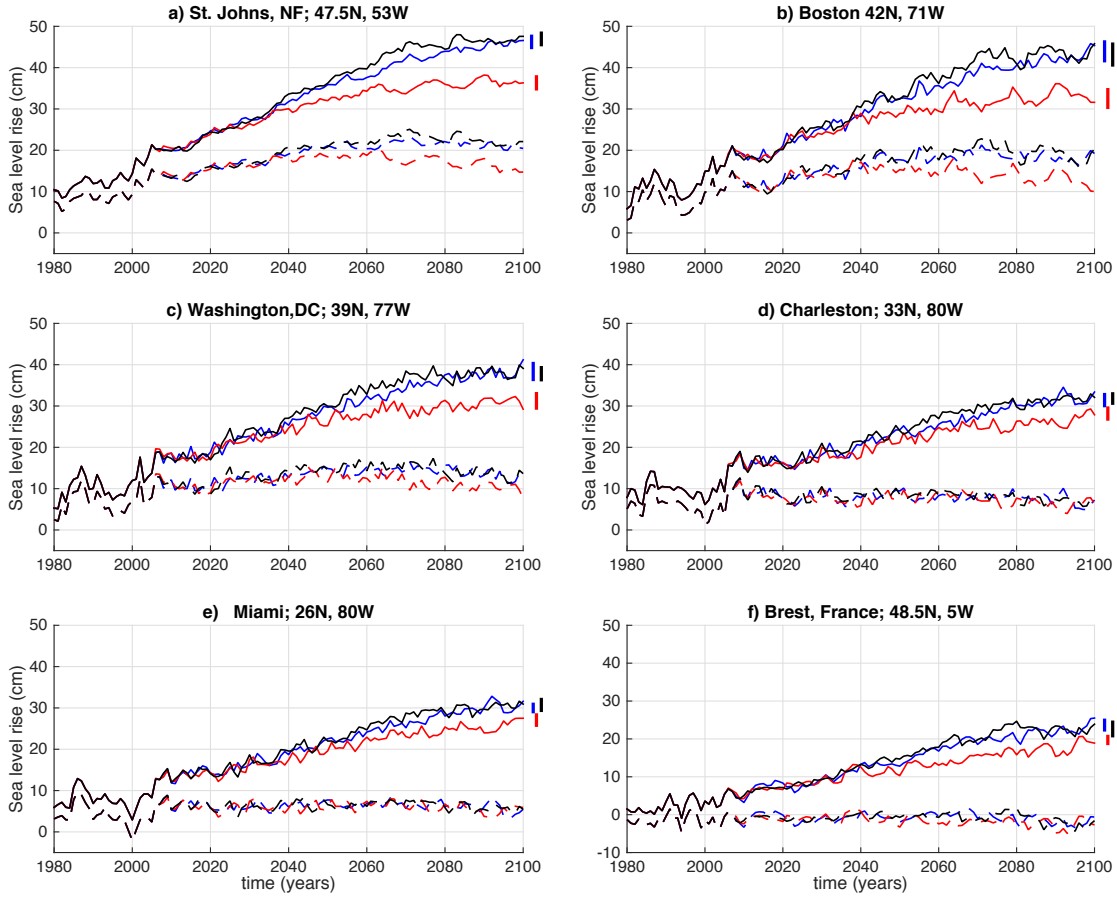

**Figure 6.** Sea level change (cm) along the North American east coast and in Brest, France. Red lines represent the stabilization pathway, black for overshoot, and blue for RCP4.5. Solid lines are the total sea level change, dashed lines show the sea level with the globally averaged steric term removed, which primarily reflects ocean circulation changes. Vertical bars show the ensemble standard deviation for the final five years of each scenario (2096-2100). a) St. John's, Newfoundland, 47.5°N; b) Boston, 42°N; c) Washington DC, 39°N; d) Charleston 33°N; e) Miami, 26°N; f) Brest, France, 48.5°N.

rise is less than the global steric average, and the three scenarios have essentially indistinguishable sea level rise at this locale in 2100 (Figure 6f). The ensemble standard deviations over 2096-2100 (vertical bars at the end of the Figure 6 time series) show that the scenario-driven differences along the North American northeast coast are greater than the internal variability (i.e. in St. Johns, Boston, Washington DC where error bars do not overlap). In contrast, the overlap of the ensemble standard deviations

5   in the southern American cities and Brest, France suggests that the internal variability is greater than the scenario-driven sea level differences at the end of the century.

## 5    Conclusions

Here we used a comprehensive Earth System Model to probe the sensitivity of $21^{st}$ century climate to the $CO_2$ concentration pathway used to limit global mean surface warming to 1.5K above preindustrial temperatures. One pathway quickly stabilizes atmospheric $CO_2$ concentrations and the other surges past the 1.5K target at mid-century, relying on hypothetical carbon removal to approach the target in 2100. Remarkably, the overshoot pathway achieves essentially the same global average temperature target with over 150 GtC greater net (diagnosed) $CO_2$ emissions because carbon uptake by both the ocean and terrestrial biospheres is strengthened at higher atmospheric $CO_2$ concentrations during the ramp-up phase. However, this seeming benefit comes at a price: the overshoot pathway leads to a 18% higher global mean steric sea level rise and stronger ocean acidification (Figure 1f). Here, we also show that overshooting the cumulative emissions necessary to limit warming to 1.5K, and then removing that carbon, would also have important consequences for the hydrological cycle, ocean circulation, the regional expressions of surface warming, sea ice, and sea level. The spin-up of the hydrological cycle, as diagnosed from global mean precipitation, follows the same temporal evolution as the radiative forcing. Notably, a slowdown in the AMOC and the associated oceanic meridional transport of heat in response to the overshoot forcing causes the ocean surface to be cooler than under stabilization forcing. Over the subpolar North Atlantic and some regions of the Southern Ocean, these cooler surfaces are associated with an expansion of sea ice relative to the stabilization scenario. With global mean temperatures essentially equal by 2100 under stabilization and overshoot forcing, this implies the presence of offsetting patches of stronger surface warming, which are found mostly over land masses and in the Arctic.

The appearance of the cooling feedback provided by the sea ice that was stronger in the overshoot than the stabilization pathway is a fascinating feature of our simulations. This sea ice expansion suggests the possibility that the time-dependent radiative feedback may strengthen during variable forcing, contrary to the common assumption that feedbacks weaken at higher temperature. The ESM2M used in our study has a relatively strong AMOC in its preindustrial control simulation and strong decline under global warming among CMIP5 models (Wang et al., 2014; Cheng et al., 2013). It also has a weaker transient climate response than most CMIP5 models (Flato et al., 2013; Gillett et al., 2013). Therefore, it is not clear how robust these feedbacks are likely to be across an ensemble of various climate models with slightly different physics.

Finally, geographic patterns of sea level rise are shown to be sensitive to the pathway to 1.5K. Due to shifting circulation in the North Atlantic, likely linked to AMOC decline, the east coast of North America could experience extremely different outcomes with respect to local sea level rise depending on $CO_2$ concentration pathway. To the south of Cape Hatteras (36°N), the difference in sea level rise between the overshoot and stabilization pathways are nearly indistinguishable from each other or from the RCP4.5 scenario which has a global mean surface temperature rise of 2K in 2100. However, sea level along the coastline north of Cape Hatteras could rise up to twice the global average, and 10 cm higher than a stabilization pathway.

*Code and data availability.*    Model output and code used in our analysis can be made available upon request from the corresponding author.

*Competing interests.* The authors declare no competing interests.

*Acknowledgements.* We thank John Dunne and Vaishali Naik for helpful comments that helped improve an earlier version of this manuscript, and Stephen Griffies who offered guidance in our intepretation of the sea level diagnostics. TLF acknowledges support from the Swiss National Science foundation grant PP00P2-170687.

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
