# Peer review of "Climate, ocean circulation, and sea level changes under stabilization and overshoot pathways to 1.5K warming"

_Earth System Dynamics, 2017_

## Referee Comment (RC1) · Anonymous Referee #1 · 21 Feb 2018

Recommendation: Accepted after minor revisions

The present manuscript presents the sensitivity of some relevant parameters: sea level changes, AMOC and global mean temperature assessed in two different scenarios of CO2 emissions (stabilization and overshoot pathways) during the XXI century, both of them reaching the recommended target of 1.5 K of global surface temperature increasing by the year 2100.

The manuscript is very well written, discussed and documented, based on appropriate referenced bibliography. The manuscript shows clearly that, even though the same global temperature is reached, the followed pathway may produce significant differ-

ences (e.g. sea level rising and AMOC), leading to scenarios with worst societal impacts than others.

There are only a few points which would be worth to refer.

1 – Authors discuss only AMOC and sea level changes. However, the Earth System model runs have produced much more output namely that concerning the global budgets of water (hydrological cycle) and angular momentum and other surface properties (e.g. precipitation leading to changes in floods and droughts occurrences). The authors may add a few sentences regarding these aspects and quantify significant changes in case they are relevant.

2 – The chosen overshot pathway is not unique. There are maybe more pessimistic ones. In the discussion, the authors may discuss which variations could be considered in the overshot pathway keeping it consistent with the final temperature rising of 1.5K (e.g. initial stage and its end before the CO2 uptaking phase) and what should be the expected impacts.

3 - Line 32, pg. 2 (2°x2.5° lat -long?) Clarify.

4 - Lines 8-9, pg. 3, clarify the type of added SST perturbations in the ensemble members: standard deviation of the perturbations, Gaussian distributed?

5 - Add dz in the integral giving the steric term (Eq. 3).

6 - Line 7, Pg. 7. Acidification is not quantized. Give some numbers clarifying the differences between the stabilization and overshoot pathway scenarios.

---

## Referee Comment (RC2) · Anonymous Referee #2 · 25 Feb 2018

This study examines the climate system response to three different emissions scenarios in one climate model. One scenario (RCP4.5) reaches 2.0C of global temperature rise above pre-industrial levels at 2100, while the other two scenarios (stabilization and overshoot) aim to reach 1.5C warming via different emission pathways by 2100. The study documents the response of global mean temperature, sea level, AMOC, and sea ice.

The manuscript addresses an important topic – the effects of different pathways to 1.5C or 2.0C warming on the climate system. It well written, well organized and the figures are clear. I believe it offers several important new messages, but needs more

evidence to back these up properly. I recommend it be accepted subject to moderate revisions (no option for this in the list of recommendations).

Comments:

1. The manuscript currently documents the responses of a range of variables to do with "Climate, ocean circulation, and sea level changes" (AMOC, sea ice, surface temperature, sea level, mixed layer heat budget for the NAtl), which is fine, but it ends up feeling a little "light" or qualitative. Many of the statements in the manuscript could easily be backed up by using the model output. It seems that sea level rise is one of the main points the authors are emphasizing? Or perhaps the North Atlantic warming hole? If so, the manuscript would benefit from some reworking to solidify these aspects . For example:

- Pg 8, paragraph starting on L5: I'm not sure I fully understand the point being made about the North Atlantic warming hole. The warming hole is "stronger" (i.e., colder) in the stabilization and overshoot scenarios than in RCP4.5. However, a) the heat budget perturbation is similar for all three scenarios while b) the AMOC weakens more for the overshoot and RCP4.5 than for the stabilization. Furthermore, the authors speculate that a "slowdown in advective heat supply may suggest a role for the horizontal gyre circulations" (should be for all three scenarios since the heat budget changes are similar for all three, but not entirely clear to me)? In the end, I'm still not sure why the warming hole is stronger in the stabilization and overshoot scenarios, thought I could speculate. All the relevant pieces of information seem to be here, they just need to be linked and interpreted together. Perhaps some zoomed in figures of the North Atlantic would help, including showing the actual changes in the gyres (Sverdrup transport) or surface heat fluxes if these are indeed important.

- Pg 7, L13-14: Could the authors show the response of the total ocean heat content and TOA radiative imbalance in the three scenarios (since these determine the steric sea level rise)?
- Fig. 2 and 4 and the global mean temperature curves suggest some interesting differences in ocean heat uptake between the three scenarios. It would be nice to see what is happening, e.g., some maps of ocean heat content, to show where the ocean ends up sequestering heat.

– What are the contributions of the global steric term and local steric term to the differences in sea level rise between the three scenarios. Figure 6 suggests that both contribute. This could be quantified for the various locations.

2. It would be useful to show some measure of how strong the signals in the 2096-2100 averages are compared to the ensemble spread (even though these are small ensembles), and internal variability (from the time series in Figure 1, it is clear there is quite a lot of interannual to decadal variability in the responses).

3. To provide some context for this single-model study, it would be useful to have some idea of how GFDL-ESM2M performs relative to other CMIP models in the RCP4.5 scenario. This is done on pg 10 for the ACC, but it would be nice to see similar discussion for variables/features that are the main focus of the manuscript.

Other comments:

1. Figure 1: the text labels are quite small, especially in panel b, and the thin lines for indivdual ensemble members are too faint.

2. Equation 3 seems to be missing a dz in the integral.

3. Pg 6, L12: arises -> arise

4. Pg 7, L10-11: "... the pathway to a given forcing does not factor prominently in the annual mean, global average surface air temperature change in response to that forcing." This statement seems a bit odd to me given that the stabilization and overshoot emissions pathways were constructed with an aim of achieving 1.5C warming at 2100. Maybe it's just that I have misunderstood the description of how the scenarios were designed, in which case the authors could try to clarify this instead (pg 3 last

**ESDD**
paragraph).

5. Figure 5: What is "vertically-averaged" sea ice concentration?

6. The implications of these results for ocean acidification is mentioned in the conclusions. This seems to be a very important point. Would it be worth including this in the results, with some analysis of the model output to back this up?

---

## Referee Comment (RC3) · Anonymous Referee #3 · 28 Feb 2018

**General comments**

**Summary**

The article's main question is how the climate system reacts to different CO2 concentration pathways to a fixed global surface warming by 2100. Having RCP4.5 as a standard scenario, the authors set up two new different CO2 pathways to 1.5K warming, the "stabilization" and "overshoot" scenarios, with the goal of reflecting two extremes of hypothetical short-term and long-term climate policy. Conducting their simulations with the GFDL-ESM2M earth system model, the authors document the global climate response in the 21st century in what respects AMOC intensity and steric sea level

rise and give the climate state by 2100 of surface temperature, sea level rise and sea ice concentration. The experiment main conclusions for overshoot pathway relative to stabilization pathway by 2100, are: (i) overshoot achieves the same global average temperature target with greater cumulative carbon emissions, but, leads to stronger ocean acidification (not quantified), higher global mean steric sea level rise and lower AMOC volume transport; (ii) overshoot forcing causes the ocean surface to be cooler over the subpolar NAtlantic and some regions of the SO with associated expansion of sea ice, suggesting a negative radiative feedback; (iii) geographic patterns of sea level rise are sensitive to the selected pathway, with overshoot forcing producing up to 10 cm of additional sea level rise in some cities of the east coast of NAmerica

**Overall evaluation**

This paper addresses a very relevant scientific question with huge societal implications. Authors carry a numerical experiment with a well-tested state of the art Earth System Model, and their results are supported by suitable simulations setup and appropriated analysis methods. The document is well structured, and its language is fluent and precise. The credit given to related work is very well balanced and authors contribution is clearly indicated. The principal study findings are clearly stated and flow natural from the presented results.

**Specific comments**

**Abstract**

The abstract gives a correct summarized perspective of the different components of the paper and it can be understood without reading the remainder document. I just have a few considerations:

- because the research is based on a single model experiment, the GFDL-ESM2M model, it should be, in my opinion, explicitly declared (line 3)

- due to its importance for marine ecosystems, I would like to see the "ocean acidification" Included in the list of other climatic metrics that show important differences in response to different CO2 concentration pathways (lines 10-11)

- rephrase lines 12-13, to make clear that the overshoot relative to the stabilization simulations gives a higher global steric sea level rise and a reduced AMOC volume transport.

1. Introduction

pg 2, line 2: to be more precise, "negative emissions" designation should also include "on-site capture of CO2"

pg 2, lines 18-23: the use of RCP4.5 and RCP8.5 scenarios in the simulations, justifies, in my opinion, to quote a summary reference(s) on these scenarios (e.g. Moss et al., 2010; van Vuuren et al., 2011)

pg 2, line 24: before paper's outline, I think that will be more appropriate to give here the justification for setting up the "stabilization" and "overshoot" pathways, than later in page 3 lines 23-25 ("Our goal in setting up the two new pathways (i.e. 'stabilization' and 'overshoot') was . . .")

2 Methods

Some of the comments intend to improve the results traceability

pg 3, line 6 – What were the criterions for choosing the number of elements in each ensemble? Why five?

pg 3, line 8 – "very tiny perturbation" . . . "similar to the approach by Wittenberg et al. (2014)". Could you please be more specific? How "tiny" is this perturbation and how "similar" is to Wittenberg et al. approach?

pg 3, line 11 – "the final year of the simulation"; do you mean 2100? The simulations period was not yet clearly stated in the text.

pg 3, line 14 – "limiting atmospheric CO2 growth rates to approximately 0.25 ppm year$-1$"; please mention the period for which this growth rate is valid. (2070-2100)?

pg 3, line 20-22 – from the observation of figures 1c and the supplementary S1 (large dots), I got somewhat different values for the three scenarios: 1.92K, 1.45K and 1.52K respectively RCP4.5, stabilization and overshoot. Did I miss something?

pg 3, line 30 – "run of GFDL-ESM2M under 1% CO2 increase"; rate is missing (1% per year).

3 Global average properties

pg 6, line 23 – suggestion: add "(TCR)" just to indicate that there is a definition behind the words "transient climate response"

pg 7, line 5 – ". . . year 2086, respectively"; refer Figure 1b

pg 7, line 9 – instead of "(Figure 1c)"; refer (Figure 1a,c).

pg 7, line 24 – "AMOC reaches its lowest point in 2075 (13.6 Sv)"; it is not possible to check this value in figure 1d! It is somewhat confusing to discuss annual values in the text and to observe 10-year running average values in the figures!

4 Regional patterns tied to the ocean circulation perturbation

pg 10, line 10 – "stabilization pathway by 2100."; refer (Figure 1e).

Figure 5, caption – please clarify "vertically-averaged sea ice concentration"

pg 13, lines 1-2 –refer Figure 6a,b for Boston and St. John's, and Figure 6d,e for Charleston and Miami

pg 13, lines 5-6 –observing Figure 4 "on the other side of Atlantic", I would be a little bit more precise and replace "southern Europe" by "southern Iberian Peninsula" and "while Northern Europe . . ." by "while further north the western European coast . . ."

pg 13, line 7 – in this line, only mention Figure 4 and refer Figure 6f at the end of the

paragraph.

pg 13 - for the analysis of the sea level change results, presented in figure 6, it would be useful to have an estimation of the internal variability (ensemble spread)

5 Conclusions

A few questions for your consideration:

- why it is omitted in your document the simulations results for the distribution of salinity by 2100? The relevance of this oceanic parameter for the AMOC dynamics and for the local steric sea level term, doesn't justify its inclusion in the discussion of the presented results?

- from the model output, is it possible to quantify ocean acidification and elaborate a little more on the evolution of this oceanic parameter under the selected scenarios?

- it should be brought to the discussion in this final section, the limitations/weaknesses of the used model that can affect the quality/representability of the obtained results (e.g. no representation of interactive ice sheets or glaciers; transient climate response (TCR); ACC position; salinity anomaly over the subtropical Atlantic (Jackson et al., 2014); AMOC depth (Kostov et al.,2014); ...). You have already made some comments at the end of section 2 (pg6, lines 15-19) that can be revisited in the context of this discussion.

Technical corrections

pg 1, line 11 – insert the (AMOC) acronym and remove the full designation in line 13

pg 2, line 33 – Anderson et al. 2004 is missing in the references list

pg 3, line 16 – "reaching a peak of 537 ppm"; observing Figure 1a this value looks like a typing error (probably, should be 573 ppm)

pg 6, equation 3 – dz is missing in the integration

pg 6, line 4 – H (the bottom depth) is not declared

pg 6, line 28 and following lines – use the same units for cumulative carbon emissions in text and Figure 1b (Pg C)

pg 9, Figure 2 – latitude and longitude labels are missing

pg 11, Figure 4 – latitude and longitude labels are missing; in figure caption should be RCP4.5 minus stabilization and overshoot minus stabilization and not the other way around

pg 13, line 2 – word "under" is repeated twice

References

Anderson et al. 2004 is missing

Gasser et al 2015; journal's name is missing: Nat. Commun.; DOI link is incorrect (http://10.1038/ncomms8958)

Gillett et al 2013; list of authors is repeated

Herweijer et al 2005; DOI link is incorrect (https://doi.org/10.3402/tellusa.v57i4.14708)

Trenberth et al 2014; list of authors is repeated

---

## Author Comment (AC1) · 18 Apr 2018

**Reviewer 1**

The present manuscript presents the sensitivity of some relevant parameters: sea level changes, AMOC and global mean temperature assessed in two different scenarios of CO2 emissions (stabilization and overshoot pathways) during the XXI century, both of them reaching the recommended target of 1.5 K of global surface temperature increasing by the year 2100.

The manuscript is very well written, discussed and documented, based on appropriate referenced bibliography. The manuscript shows clearly that, even though the same global temperature is reached, the followed pathway may produce significant differences (e.g. sea level rising and AMOC), leading to scenarios with worst societal impacts than others.

We thank the reviewer for a supportive, insightful and thorough review, which helped us to improve the manuscript. We respond to each point in blue text interwoven with the reviewer comments below.  All major additions to the manuscript are also highlighted in blue text.

There are only a few points which would be worth to refer.

1 – Authors discuss only AMOC and sea level changes. However, the Earth System model runs have produced much more output namely that concerning the global budgets of water (hydrological cycle) and angular momentum and other surface properties (e.g. precipitation leading to changes in floods and droughts occurrences). The authors may add a few sentences regarding these aspects and quantify significant changes in case they are relevant.

We agree that there are many more variables that could be potentially interesting.  We remain focused on the original set of variables, but have added a few additional time series figures.  The first was pH, which was requested by all three reviewers.  It now appears as a panel in Figure 1, with relevant discussion and quantification in Section 3.  In addition, we have added a Supplementary Figure of global average precipitation, with the following brief discussion in Section 3:

Because the hydrological cycle tends to spin-up as the climate warms (Held and Soden, 2006) the global average precipitation increases approximately 1% over the 21st century in all three scenarios.  The rate of change seems to roughly track the global temperature change, therefore leading to slightly stronger increases in Overshoot and RCP4.5 starting at mid-century (Supplementary Figure 2).  However, internal variability causes substantial overlap between ensemble members.

Finally, we also explored the salinity in the subpolar North Atlantic, since a slowdown of the AMOC is central to the manuscript, and is often assumed to be triggered by salinity changes. We found that subpolar North Atlantic salinity declined in all three models, but declined fastest in the Overshoot ensemble, as now detailed towards the beginning of Section 4.

2 – The chosen overshot pathway is not unique. There are maybe more pessimistic ones. In the discussion, the authors may discuss which variations could be considered in the overshot pathway keeping it consistent with the final temperature rising of 1.5K (e.g. initial stage and its end before the CO2 uptaking phase) and what should be the expected impacts.

We now reference other studies that have compared various overshoot simulations of different intensity during ramp-up and ramp-down, as well timing of the inflection point in the Methods on page 3.

3 - Line 32, pg. 2 (2 x2.5°) lat -long?) Clarify.
Yes, the reviewer is correct in that the resolution is 2° latitude by 2.5° longitude, and the clarification has been added to the manuscript

4 - Lines 8-9, pg. 3, clarify the type of added SST perturbations in the ensemble members: standard deviation of the perturbations, Gaussian distributed?
For each ensemble member $i = 1,2 \ldots, 5$, we added: $dT = 0.0001°C *i$ to a single ocean grid cell in the Weddell Sea at 5-m depth. This information has been added to the Methods.

5 - Add dz in the integral giving the steric term (Eq. 3).
Thank you for catching this omission. It has been corrected.

6 - Line 7, Pg. 7. Acidification is not quantized. Give some numbers clarifying the differences between the stabilization and overshoot pathway scenarios
We have added a panel to Figure 1 to show the pH changes over time in the surface layer, and a deeper layer (1573 m). We also added the following text to the manuscript to describe the changes:

The higher atmospheric carbon concentrations in the overshoot pathway relative to the stabilization pathway temporarily cause more ocean acidification (i.e. lower pH values) at the surface (Fig. 1f). In 2100, when the atmospheric $CO_2$ in overshoot nearly matches stabilization, the surface pH is still lower in overshoot, but is rapidly approaching the stabilization surface value. At depth (e.g. 1573 m), both the emergence of greater acidification as well as the turn-around in pH in the overshoot scenario are delayed by about 20 years relative to the trends in atmospheric $CO_2$, as transport and mixing between the surface and interior ocean is relatively slow (Gebbie 2012). As a result, ocean pH is lower in the overshoot scenario than in the stabilization scenario at the end of the 21st century, as expected from the carbon budget (Figure 1b). Based on the rate of change at 2100 in our simulations, we expect this difference in pH to persist for a few decades after the crossover point in atmospheric $CO_2$, as the ocean slowly releases the excess $CO_2$ to the atmosphere.

---

## Author Comment (AC2) · 18 Apr 2018

**Reviewer 2**
This study examines the climate system response to three different emissions scenarios in one climate model. One scenario (RCP4.5) reaches 2.0C of global temperature rise above pre-industrial levels at 2100, while the other two scenarios (stabilization and overshoot) aim to reach 1.5C warming via different emission pathways by 2100. The study documents the response of global mean temperature, sea level, AMOC, and sea ice.

The manuscript addresses an important topic – the effects of different pathways to 1.5C or 2.0C warming on the climate system. It well written, well organized and the figures are clear. I believe it offers several important new messages, but needs more evidence to back these up properly. I recommend it be accepted subject to moderate revisions (no option for this in the list of recommendations).
We thank the reviewer for a supportive, insightful and thorough review, which helped us to improve the manuscript. We respond to each point in blue text interwoven with the reviewer comments below.  All major additions to the manuscript are also highlighted in blue text.

Comments:
1. The manuscript currently documents the responses of a range of variables to do with "Climate, ocean circulation, and sea level changes" (AMOC, sea ice, surface temperature sea level, mixed layer heat budget for the NAtl), which is fine, but it ends up feeling a little "light" or qualitative. Many of the statements in the manuscript could easily be backed up by using the model output. It seems that sea level rise is one of the main points the authors are emphasizing? Or perhaps the North Atlantic warming hole? If so, the manuscript would benefit from some reworking to solidify these aspects. For example:
We have added additional quantifications to the text and have reworked aspects of the text. Details follow below in response to specific suggestions.

– Pg 8, paragraph starting on L5: I'm not sure I fully understand the point being made about the North Atlantic warming hole. The warming hole is "stronger" (i.e., colder) in the stabilization and overshoot scenarios than in RCP4.5. However, a) the heat budget perturbation is similar for all three scenarios while b) the AMOC weakens more for the overshoot and RCP4.5 than for the stabilization. Furthermore, the authors speculate that a "slowdown in advective heat supply may suggest a role for the horizontal gyre circulations" (should be for all three scenarios since the heat budget changes are similar for all three, but not entirely clear to me)? In the end, I'm still not sure why the warming hole is stronger in the stabilization and overshoot scenarios, though I could speculate. All the relevant pieces of information seem to be here, they just need to be linked and interpreted together. Perhaps some zoomed in figures of the North Atlantic would help, including showing the actual changes in the gyres (Sverdrup transport) or surface heat fluxes if these are indeed important.

The warming hole is not substantially different between the different scenarios, as we try to emphasize more clearly in the revised text. Figure 3 is meant to show that the temperature tendency is the result of a very small residual of large, compensating physical processes: namely, the advective supply of heat and its loss through air-sea exchange, with a subdominant contribution from mixing of heat into the region.  As the circulation changes, both the advective supply of heat and the air-sea exchange of heat is reduced by more than 75% in both scenarios,

while the mixing supply of heat increases slightly. The ultimate rate of cooling is determined by the balance of these processes, with the cooling due to the slowdown in advective heat supply "winning" by a very small margin. The balance of these processes is quite similar among the models and we did not mean to emphasize the slightly cooler temperatures. Rather, our intention was to find a causal explanation for the cool region, since the cause of subpolar temperature anomalies is a source of much debate [*Clement et al.*, 2015; *Bellomo et al.*, 2017; *Zhang*, 2017; *Caesar et al.*, 2018]. We agree that the breakdown of the heat transport into horizontal gyre versus overturning circulation is another interesting direction, but would argue that this is a separate line of inquiry, that has been recently explored [*Piecuch et al.*, 2017]. The decomposition here must be done in isopycnal space since isopycnals slope both zonally and meridionally in the subpolar North Atlantic. So, the horizontal transport can cross isopycnals and be part of the overturning circulation, as defined in density space.

- Pg 7, L13-14: Could the authors show the response of the total ocean heat content and TOA radiative imbalance in the three scenarios (since these determine the steric sea level rise)? Figure 1 of this response document (below) gives the total ocean heat content change (in joules) over the 21$^{st}$ century, which is essentially the same as the time integral of the TOA radiative imbalance, given the small heat capacity of the atmosphere and the lack of interactive ice sheets in the model. These curves show almost precisely the same pattern as the global steric sea level rise, since the lion's share of steric sea level rise is due to the thermosteric effect. Even though the equation of state is non-linear, there is nothing notable to distinguish this time series from the steric sea level (Figure 1e). Therefore, we have not added this figure, but note the similarity in the time evolution in the text (lines 22-25 on page 7).

[Figure]

Figure 1: Global ocean heat content anomaly (relative to 2006) for the ensemble mean of Stabilization (red), RCP4.5 (blue), and Overshoot (black) scenarios.

– Fig. 2 and 4 and the global mean temperature curves suggest some interesting differences in

ocean heat uptake between the three scenarios. It would be nice to see what is happening, e.g., some maps of ocean heat content, to show where the ocean ends up sequestering heat.

We now provide heat uptake maps in the Supplementary information, with a brief discussion on page 7: "As has been noted in previous studies with this model, the heat uptake per area in the Atlantic outpaces the other basins, with this pattern arising in part due to the overturning circulation and its slowdown [e.g. *Winton et al.*, 2013]. Notably, the Atlantic heat uptake in RCP4.5 and Overshoot are slightly faster than the uptake in Stabilization, even after removing the global mean uptake rate."

– What are the contributions of the global steric term and local steric term to the differences in sea level rise between the three scenarios. Figure 6 suggests that both contribute. This could be quantified for the various locations.

Figure 6 shows the total local sea level rise (solid line), which is the sum of the local steric effect, the global steric effect, and the local change in ocean mass (due to surface fluxes and column-integrated mass convergence). The dashed lines show just the sum of the two local terms, without the global steric effect. We believe this decomposition gives a good indication of the changing ocean dynamics that leads to local sea level differences. Further decomposition of the local term into components might not be as revealing as one would hope, since the local steric effect will be influenced by dynamical shifts that transport water of different temperatures around the ocean. Further, it is not trivial to meaningfully separate the various terms in the sea level equation (see Griffies and Greatbach [2012] for a thorough exploration of sea level in ocean models). Specifically, we could further separate the local term into local steric effect and the local change in mass, but then we would have to correct for a spurious mass source that arises in Boussinesq models (see Appendix D of Griffies and Greatbach [2012]). For these reasons, we continue to rely on Figure 6 to explain the sea level differences among the scenarios.

2. It would be useful to show some measure of how strong the signals in the 2096-2100 averages are compared to the ensemble spread (even though these are small ensembles), and internal variability (from the time series in Figure 1, it is clear there is quite a lot of interannual to decadal variability in the responses).

Thank you for this suggestion. We have added the ensemble standard deviation for the final 5 years (2096-2100) to Figure 6, which makes our point more clearly that the stabilization and overshoot pathways can create large differences in some regions (Washington, Boston, St. Johns) and almost no difference in others (Charleston, Miami, Brest). In the latter three cities, the ensemble standard deviations overlap, an indication that internal variability is larger than the scenario-driven difference in sea level at these locations.

3. To provide some context for this single-model study, it would be useful to have some idea of how GFDL-ESM2M performs relative to other CMIP models in the RCP4.5 scenario. This is done on pg 10 for the ACC, but it would be nice to see similar discussion for variables/features that are the main focus of the manuscript.

We have added brief discussions of the following points, aimed at contextualizing GFDL-ESM2M relative to other CMIP models:

-   Bottom of page 6: We note that the transient climate response in GFDL ESM2M is at the lower end of the CMIP5 model suite.

- Middle of page 7: Model differences in surface pH changes are small [*Frölicher et al.*, 2016], so we would expect that the surface pH evolution to be robust across models.
- Top of page 8: The decrease in AMOC is relatively large in comparison with other models, because mean state is also at the higher end.

Other comments:
1. Figure 1: the text labels are quite small, especially in panel b, and the thin lines for individual ensemble members are too faint.
The text labels have been enlarged. The ensemble members in panel b are just very close together, and therefore hard to distinguish. They are clearer in other panels, in which the ensemble variability is greater.

2. Equation 3 seems to be missing a dz in the integral.
Thank you for catching this omission. We corrected it.

3. Pg 6, L12: arises -> arise
Fixed.

4. Pg 7, L10-11: "... the pathway to a given forcing does not factor prominently in the annual mean, global average surface air temperature change in response to that forcing." This statement seems a bit odd to me given that the stabilization and overshoot emissions pathways were constructed with an aim of achieving 1.5C warming at 2100. Maybe it's just that I have misunderstood the description of how the scenarios were designed, in which case the authors could try to clarify this instead (pg 3 last paragraph).
It is very difficult to construct scenarios in an AOGCM that match exactly the desired temperature in a given year. Thus, this statement was meant to highlight the success of the method used to construct the pathways, as well as make clear that the nonlinearities that arise under these scenarios do not cause major deviations from the expected global mean temperature pathway.

5. Figure 5: What is "vertically-averaged" sea ice concentration?
There are 5 vertical layers of sea ice. The concentration over all five layers sums to one if the grid cell is covered in sea ice for the whole year over all five vertical layers. We agree that "vertically-averaged" sea ice concentration is confusing, and therefore changed this graphic in the final version and its corresponding description. The revised figure shows the change in vertical sum of the concentrations (with a maximum value of 1) between the preindustrial 2096-2100 year average of the ensemble mean and the preindustrial control.

6. The implications of these results for ocean acidification is mentioned in the conclusions. This seems to be a very important point. Would it be worth including this in the results, with some analysis of the model output to back this up?
As in response to Reviewer 1: We added a panel to Figure 1 to show the pH changes over time in the surface layer, and a deeper layer (1573 m). We also added the following text to the manuscript to describe the changes:

The higher atmospheric carbon concentrations in the overshoot pathway relative to the

stabilization pathway temporarily cause more ocean acidification (i.e. lower pH values) at the surface (Fig. 1f). At the point that atmospheric $CO_2$ in overshoot matches nearly stabilization (2100), the surface pH is still lower in overshoot, but is rapidly approaching the stabilization surface value. At depth (i.e. 1573m), both the emergence of greater acidification as well as the turn-around in pH in the overshoot scenario are delayed by about 20 years relative to the trends in atmospheric CO2, because transport and mixing between the surface and depth is relatively slow [Gebbie et al., 2012]. As a result, ocean pH is lower in the overshoot scenario than in the stabilization scenario at the end of the 21st century, as expected from the carbon budget (Fig. 1b). Based on the rate of change at 2100 in our simulations, we expect this difference in pH to persist for a few decades after the crossover point in atmospheric $CO_2$, as the ocean slowly releases the excess $CO_2$ to the atmosphere

---

## Author Comment (AC3) · 18 Apr 2018

**Reviewer 3**

**General comments**
**Summary**
The article's main question is how the climate system reacts to different CO2 concentration pathways to a fixed global surface warming by 2100. Having RCP4.5 as a standard scenario, the authors set up two new different CO2 pathways to 1.5K warming, the "stabilization" and "overshoot" scenarios, with the goal of reflecting two extremes of hypothetical short-term and long-term climate policy. Conducting their simulations with the GFDL-ESM2M earth system model, the authors document the global climate response in the 21st century in what respects AMOC intensity and steric sea level rise and give the climate state by 2100 of surface temperature, sea level rise and sea ice concentration. The experiment main conclusions for overshoot pathway relative to stabilization pathway by 2100, are: (i) overshoot achieves the same global average temperature target with greater cumulative carbon emissions, but, leads to stronger ocean acidification (not quantified), higher global mean steric sea level rise and lower AMOC volume transport; (ii) overshoot forcing causes the ocean surface to be cooler over the subpolar NAtlantic and some regions of the SO with associated expansion of sea ice, suggesting a negative radiative feedback; (iii) geographic patterns of sea level rise are sensitive to the selected pathway, with overshoot forcing producing up to 10 cm of additional sea level rise in some cities of the east coast of NAmerica

We thank the reviewer for a supportive, insightful and thorough review, which helped us to improve the manuscript. We respond to each point in blue text interwoven with the reviewer comments below.  All major additions to the manuscript are also highlighted in blue text.

**Overall evaluation**
This paper addresses a very relevant scientific question with huge societal implications. Authors carry a numerical experiment with a well-tested state of the art Earth System Model, and their results are supported by suitable simulations setup and appropriated analysis methods. The document is well structured, and its language is fluent and precise. The credit given to related work is very well balanced and authors contribution is clearly indicated. The principal study findings are clearly stated and flow natural from the presented results.

**Specific comments**
Abstract
The abstract gives a correct summarized perspective of the different components of the paper and it can be understood without reading the remainder document. I just have a few considerations:
  - Because the research is based on a single model experiment, the GFDL-ESM2M model, it should be, in my opinion, explicitly declared (line 3)

The model is now specified in the abstract.

  - Due to its importance for marine ecosystems, I would like to see the "ocean acidification" Included in the list of other climatic metrics that show important differences in response to different CO2 concentration pathways (lines 10-11)

We have added ocean acidification to this list.

• rephrase lines 12-13, to make clear that the overshoot relative to the stabilization simulations gives a higher global steric sea level rise and a reduced AMOC volume transport.

Clarified.

1. Introduction
• pg 2, line 2: to be more precise, "negative emissions" designation should also include "on-site capture of CO2"

We have added reference to on-site capture.

• pg 2, lines 18-23: the use of RCP4.5 and RCP8.5 scenarios in the simulations, justifies, in my opinion, to quote a summary reference(s) on these scenarios (e.g. Moss et al., 2010; van Vuuren et al., 2011)

These references have been added.

• pg 2, line 24: before paper's outline, I think that will be more appropriate to give here the justification for setting up the "stabilization" and "overshoot" pathways, than later in page 3 lines 23-25 ("Our goal in setting up the two new pathways (i.e. 'stabilization' and 'overshoot') was : : :")

We have reorganized this text as suggested.

2 Methods
Some of the comments intend to improve the results traceability
• pg 3, line 6 – What were the criterions for choosing the number of elements in each ensemble? Why five?

We added the following text to the Methods: Five ensemble members provide a means of averaging out internal variability in order to more clearly separate the differences in the simulations arising from the $CO_2$ forcing at reasonable computational expense.

• pg 3, line 8 – "very tiny perturbation" : : : "similar to the approach by Wittenberg et al. (2014)". Could you please be more specific? How "tiny" is this perturbation and how "similar" is to Wittenberg et al. approach?

For each ensemble member $i = 1,2 \ldots, 5$, we added:  $dT = 0.0001°C *i$  to a single ocean grid cell in the Weddell Sea at 5-m depth.  This information has been added to the Methods.

• pg 3, line 11 – "the final year of the simulation"; do you mean 2100? The simulations period was not yet clearly stated in the text.

Yes, 2100, as clarified in the revised text.

• pg 3, line 14 – "limiting atmospheric CO2 growth rates to approximately 0.25 ppm year[1]"; please mention the period for which this growth rate is valid. (2070-2100)

The following details were added to the Methods:
    This warming target was achieved by setting atmospheric $CO_2$ growth rates to be approximately 0.65 ppm year[-1] from 2020 to 2060, and then having these growth rates decline to nearly zero by the end of the century.  Under these slow growth rates, atmospheric $CO_2$

concentrations never exceed 435 ppm before 2100 (Figure 1a).

- pg 3, line 20-22 – from the observation of figures 1c and the supplementary S1 (large dots), I got somewhat different values for the three scenarios: 1.92K, 1.45K and 1.52K respectively RCP4.5, stabilization and overshoot. Did I miss something?

The numbers given in the text were annual means taken in 2100. The figures show 5-year running means. We have made an additional note about this subtlety on page 3, line 24:

"Here, we give the ensemble mean temperature in 2100, since this is the year that the $CO_2$ forcing is approximately equal in the stabilization and overshoot simulations. The ensemble averages over the years 2096-2100, which are reflected in the final year of the smoothed time series in Figure 1c, are slightly different (1.95 ± 0.05K in RCP4.5; 1.48 ± 0.09K in stabilization; and 1.56 ± 0.09K in overshoot)."

- pg 3, line 30 – "run of GFDL-ESM2M under 1% CO2 increase"; rate is missing (1% per year).

Fixed, thank you.

3 Global average properties
- pg 6, line 23 – suggestion: add "(TCR)" just to indicate that there is a definition behind the words "transient climate response"

Added.

- pg 7, line 5 – ": : : year 2086, respectively"; refer Figure 1

Added.

- pg 7, line 9 – instead of "(Figure 1c)"; refer (Figure 1a,c).

Added

- pg 7, line 24 – "AMOC reaches its lowest point in 2075 (13.6 Sv)"; it is not possible to check this value in figure 1d! It is somewhat confusing to discuss annual values in the text and to observe 10-year running average values in the figures!

We apologize for this confusion. We have replaced the annual means in the text with the 10-year means centered at 2075, which is 14.6 Sv and can be visually matched with Figure 1d.

4 Regional patterns tied to the ocean circulation perturbation
- pg 10, line 10 – "stabilization pathway by 2100."; refer (Figure 1e).

Added.

- Figure 5, caption – please clarify "vertically-averaged sea ice concentration"

We agree that "vertically-averaged" sea ice concentration is confusing, and therefore changed this graphic in the final version. There are 5 vertical layers of sea ice. The concentration over all five layers sums to one if the grid cell is covered in sea ice for the whole year over all five vertical layers. Therefore, the revised figure shows the sum of the concentrations over the five levels and the change in that sum between the preindustrial and the final five year average of the

ensemble mean.

- pg 13, lines 1-2 –refer Figure 6a,b for Boston and St. John's, and Figure 6d,e for Charleston and Miami

Added.

- pg 13, lines 5-6 –observing Figure 4 "on the other side of Atlantic", I would be a little bit more precise and replace "southern Europe" by "southern Iberian Peninsula" and "while Northern Europe : : :" by "while further north the western European coast : : :"

We have made the suggested changes.

- pg 13, line 7 – in this line, only mention Figure 4 and refer Figure 6f at the end of the paragraph.

Done.

- pg 13 - for the analysis of the sea level change results, presented in figure 6, it would be useful to have an estimation of the internal variability (ensemble spread)

We have revised the figure to include ensemble standard deviations.

5 Conclusions

A few questions for your consideration:
- Why it is omitted in your document the simulations results for the distribution of salinity by 2100? The relevance of this oceanic parameter for the AMOC dynamics and for the local steric sea level term, doesn't justify its inclusion in the discussion of the presented results?

We have added a brief discussion of the salinity change in the subpolar North Atlantic, as this is the region most relevant for AMOC dynamics.  The halosteric term typically makes a very small contribution to global sea level [*Griffies et al.*, 2014; *Palter et al.*, 2014].

- From the model output, is it possible to quantify ocean acidification and elaborate a little more on the evolution of this oceanic parameter under the selected scenarios?

Yes, we have added a panel to Figure 1 and some additional text to make this result more quantitative.

- It should be brought to the discussion in this final section, the limitations/weaknesses of the used model that can affect the quality/representability of the obtained results (e.g. no representation of interactive ice sheets or glaciers; transient climate response (TCR); ACC position; salinity anomaly over the subtropical Atlantic (Jackson et al., 2014); AMOC depth (Kostov et al.,2014). You have already made some comments at the end of section 2 (pg6, lines 15-19) that can be revisited in the context of this discussion.

We have inserted a few sentences on page 15 (line 17-20) to contextualize this model among other CMIP5 models with respect to the key points emphasized in the conclusions.

Technical corrections
- pg 1, line 11 – insert the (AMOC) acronym and remove the full designation in line 13
- pg 2, line 33 – Anderson et al. 2004 is missing in the references list

- pg 3, line 16 – "reaching a peak of 537 ppm"; observing Figure 1a this value looks like a typing error (probably, should be 573 ppm)
- pg 6, equation 3 – dz is missing in the integration

All corrected. We appreciate your attention to these details.

---

## Author Comment (AC4) · 18 Apr 2018

[revised manuscript text omitted]

---

## Author Response (AR1)

The note from the editor indicated that the response to the reviewers and marked up manuscript submitted in the interactive discussion was sufficient to move on toward submission of the final documents. Therefore, we have uploaded the revised manuscript, abstract and supplement here, but have not recreated what we did in the previous step.